# CONTINUAL LEARNING USING HASH-ROUTED CONVOLUTIONAL NEURAL NETWORKS

## ABSTRACT

Continual learning could shift the machine learning paradigm from *data centric* to *model centric*. A continual learning model needs to scale efficiently to handle semantically different datasets, while avoiding unnecessary growth. We introduce hash-routed convolutional neural networks: a group of convolutional units where data flows dynamically. Feature maps are compared using feature hashing and similar data is routed to the same units. A hash-routed network provides excellent plasticity thanks to its routed nature, while generating stable features through the use of orthogonal feature hashing. Each unit evolves separately and new units can be added (to be used only when necessary). Hash-routed networks achieve excellent performance across a variety of typical continual learning benchmarks without storing raw data and train using only gradient descent. Besides providing a continual learning framework for supervised tasks with encouraging results, our model can be used for unsupervised or reinforcement learning.

## 1 INTRODUCTION

When faced with a new modeling challenge, a data scientist will typically train a model from a class of models based on her/his expert knowledge and retain the best performing one. The trained model is often useless when faced with different data. Retraining it on new data will result in poor performance when trying to reuse the model on the original data. This is what is known as *catastrophic forgetting* (McCloskey & Cohen, 1989). Although transfer learning avoids retraining networks from scratch, keeping the acquired knowledge in a trained model and using it to learn new tasks is not straightforward. The real knowledge remains with the human expert. Model training is usually a *data centric* task. Continual learning (Thrun, 1995) makes model training a *model centric* task by maintaining acquired knowledge in previous learning tasks.

Recent work in continual (or lifelong) learning has focused on supervised classification tasks and most of the developed algorithms do not generate stable features that could be used for unsupervised learning tasks, as would a more generic algorithm such as the one we present. Models should also be able to adapt and scale reasonably to accommodate different learning tasks without using an exponential amount of resources, and preferably with little data scientist intervention.

To tackle this challenge, we introduce hash-routed networks (HRN). A HRN is composed of multiple independent processing units. Unlike typical convolutional neural networks (CNN), the data flow between these units is determined dynamically by measuring similarity between hashed feature maps. The generated feature maps are stable. Scalability is insured through unit evolution and by increasing the number of available units, while avoiding exponential memory use.

This new type of network maintains stable performance across a variety of tasks (including semantically different tasks). We describe expansion, update and regularization algorithms for continual learning. We validate our approach using multiple publicly available datasets, by comparing supervised classification performance. Benchmarks include Pairwise-MNIST, MNIST/Fashion-MNIST (Xiao et al., 2017) and SVHN/incremental-Cifar100 (Netzer et al., 2011; Krizhevsky et al., 2009).

Relevant background is introduced in section 2. Section 3 details the hash-routing algorithm and discusses its key attributes. Section 4 compares our work with other continual learning and dynamic network studies. A large set of experiments is carried out in section 5.

## 2 FEATURE HASHING BACKGROUND

Feature hashing, also known as the *hashing trick* (Weinberger et al., 2009) is a dimension reduction transformation with key properties for our work: inner product conservation and quasi-orthogonality. A feature hashing function $\phi : \mathbb{R}^N \to \mathbb{R}^s$, can be built using two uniform hash functions $h : \mathbb{N} \to \{1, 2..., s\}$ and $\xi : \mathbb{N} \to \{-1, 1\}$, as such:

$$\phi_i(\mathbf{x}) = \sum_{\substack{j \in [[1,N]] \\ j : h(j) = i}} \xi(j) x_j$$

where $\phi_i$ denotes the $i^{th}$ component of $\phi$. Inner product is preserved as $\mathbb{E}[\phi(\mathbf{a})^T \phi(\mathbf{b})] = \mathbf{a}^T \mathbf{b}$. $\phi$ provides an unbiased estimator of the inner product. It can also be shown that if $||\mathbf{a}||_2 = ||\mathbf{b}||_2 = 1$, then $\sigma_{\mathbf{a},\mathbf{b}} = \mathcal{O}(\frac{1}{s})$.

Two different hash functions $\phi$ and $\phi'$ (e.g. $h \neq h'$ or $\xi \neq \xi'$) are orthogonal. In other words, $\forall (\mathbf{v}, \mathbf{w}) \in Im(\phi) \times Im(\phi'), \mathbb{E}[\mathbf{v}^T \mathbf{w}] \approx 0$. Furthermore, Weinberger et al. (2009) details the inner product bounds, given $\mathbf{v} \in Im(\phi)$ and $\mathbf{x} \in \mathbb{R}^N$:

$$Pr(|\mathbf{v}^T \phi'(\mathbf{x})| > \epsilon) \leq 2 \exp\left(-\frac{\epsilon^2/2}{s^{-1} \|\mathbf{v}\|_2^2 \|\mathbf{x}\|_2^2 + \|\mathbf{v}\|_\infty \|\mathbf{x}\|_\infty \epsilon/3}\right) \tag{1}$$

Eq.1 shows that approximate orthogonality is better when $\phi'$ handles bounded vectors. Data independent bounds can be obtained by setting $\|\mathbf{x}\|_\infty = 1$ and replacing $\mathbf{v}$ by $\frac{\mathbf{v}}{\|\mathbf{v}\|_2}$, which leads to $\|\mathbf{x}\|_2^2 \leq N$ and $\|\mathbf{v}\|_\infty \leq 1$, hence:

$$Pr(|\mathbf{v}^T \phi'(\mathbf{x})| > \epsilon) \leq 2 \exp\left(-\frac{\epsilon^2/2}{s^{-1} \|\mathbf{x}\|_2^2 + \|\mathbf{v}\|_\infty \epsilon/3}\right) \leq 2 \exp\left(-\frac{\epsilon^2/2}{N/s + \epsilon/3}\right) \tag{2}$$

Better approximate orthogonality significantly reduces correlation when summing feature vectors generated by different hashing functions, as is done in hash-routed networks.

## 3 HASH-ROUTED NETWORKS

### 3.1 STRUCTURE

A hash-routed network maps input data to a feature vector of size $s$ that is stable across successive learning tasks. An HRN exploits inner product preservation to insure that similarity between generated feature vectors reflect the similarity between input samples. Quasi-orthogonality of different feature hashing functions is used to reduce correlation between the output's components, as it is the sum of individual hashed feature vectors. An HRN $\mathcal{H}$ is composed of $M$ units $\{\mathcal{U}_1, ..., \mathcal{U}_M\}$. Each unit $\mathcal{U}_k$ is composed of:

- A series of convolution operations $f_k$. It is characterized by a number of input channels and a number of output channels, resulting in a vector of trainable parameters $\mathbf{w}_k$. Note that $f_k$ can also include pooling operations.
- An orthonormal projection basis $\mathbf{B}_k$. It contains a maximum of $m$ non-zeros orthogonal vectors of size $s$. Each basis is filled with zero vectors at first. These will be replaced by non-zero vectors during training.
- A feature hashing function $\phi_k$ that maps a feature vector of any size to a vector of size $s$.

The network also has an independent feature hashing function $\phi_0$. All the feature hashing functions are different but generate feature vectors of size $s$.

### 3.2 OPERATION

#### 3.2.1 HASH-ROUTING ALGORITHM

$\mathcal{H}$ maps an input sample $x$ to a feature vector $\mathcal{H}(x)$ of size $s$. In a vanilla CNN, $x$ would go through a series of deterministic convolutional layers to generate feature maps of growing size. In a HRN,

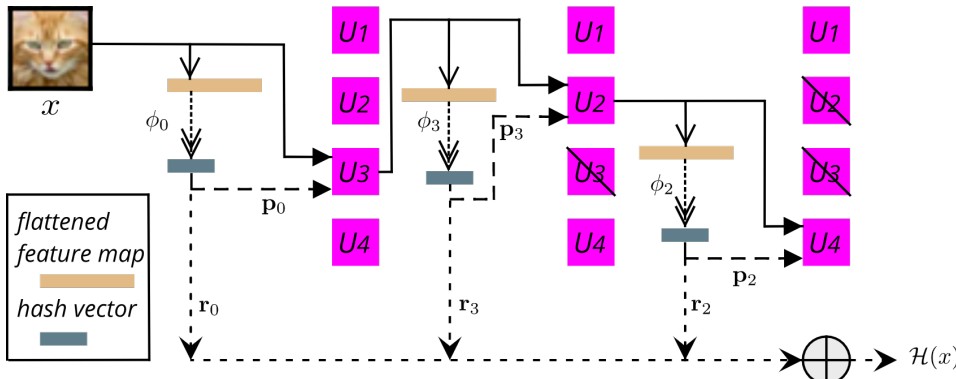

Figure 1: A hash-routed network with 4 units and a depth of 3. In this example, $\mathcal{U}_3$ is selected first as the hashed flattened image has the highest projection ($\mathbf{p}_0$) magnitude onto its basis. The structured image passes through the unit's convolution filters, generating the feature map in the middle. This process is repeated twice whilst disregarding used units at each level. The final output is the sum of all projection residues. *Best viewed in color.*

the convolutional layers that will be involved will vary depending on intermediate results.

Feature hashing is used to route operations. Since feature hashing preserves the inner product in the hashed features space, similar samples will be processed similarly. Intermediate features are hashed and projected upon the units' projection bases. The unit where the projection's magnitude is the highest is selected for the next operation. Operations continue until a maximum depth $d$ is reached (i.e. there is a limit of $d - 1$ chained operations), or when the projection residue is below a given threshold $\tau_d$. $\mathcal{H}(x)$ is the sum of all residues.

Let $\{\mathcal{U}_{i_1}, \mathcal{U}_{i_2}, ..., \mathcal{U}_{i_{d-1}}\}$ be the ordered set of units involved in processing $x$ (assuming the final projection residue's magnitude is greater than $\tau_d$). Operation 0 simply involves hashing the (flattened) input sample using $\phi_0$. Let $x_{i_k} = f_{i_k} \circ f_{i_{k-1}} \circ ... \circ f_{i_1}(x)$ be the intermediate features obtained at operation $k$. The normalized hashed features vector after operation $k$ is computed as such:

$$\mathbf{h}_{i_k} = \frac{\phi_{i_k}\left(\frac{x_{i_k}}{\|x_{i_k}\|_\infty}\right)}{\left\|\phi_{i_k}\left(\frac{x_{i_k}}{\|x_{i_k}\|_\infty}\right)\right\|_2} \tag{3}$$

For operation 0, $\mathbf{h}_{i_0}$ is computed using $x$ and $\phi_0$.

$\mathbf{p}_{i_k} = \mathbf{B}_{i_{k+1}}\mathbf{h}_{i_k}$ and $\mathbf{r}_{i_k} = \mathbf{h}_{i_k} - \mathbf{p}_{i_k}$ are the projection vector and residue vector over basis $\mathbf{B}_{i_{k+1}}$ *resp*. As explained earlier, this means that:

$$i_{k+1} = \underset{j \in \mathcal{I}\setminus\{i_1,...,i_k\}}{\arg\max} \|\mathbf{B}_j\mathbf{h}_{i_k}\|_2 \tag{4}$$

where $\mathcal{I}$ is the subset of initialized units (i.e. units with bases containing at least one non-zero vector). Finally,

$$\mathcal{H}(x) = \sum_{j \in \{i_0,...,i_{d-1}\}} \mathbf{r}_j \tag{5}$$

The full inference algorithm is summarized in Algorithm 1 and an example is given in Figure 1.

### 3.2.2 ANALYSIS

The output of a typical CNN is a feature map with a dimension that depends on the number of output channels used in each convolutional layer. In a HRN, this would lead to a variable dimension output as the final feature map depends on the routing. In a continual learning setup, dealing with variable dimension feature maps would be impractical. Feature hashing circumvents these problems by generating feature vectors of fixed dimension.

Similar feature maps get to be processed by the same units, as a consequence of using feature hashing

---

**Algorithm 1:** Hash-routed inference

---

**Input:** $x$
**Output:** $H = \mathcal{H}(x)$
$\mathbf{h_0} = \phi_0(x); \mathcal{J} = \emptyset$
$H \leftarrow 0; \mathbf{h} \leftarrow \mathbf{h_0}; y \leftarrow x$
**for** $j = 1, ..., d-1$ **do**
    $i_j = \arg\max_{k \in \mathcal{I} \setminus \mathcal{J}} \|\mathbf{B}_k \mathbf{h}\|_2$ ;
      // select the best unit
    $\mathbf{r} \leftarrow \mathbf{h} - \mathbf{B}_{i_j}\mathbf{h}$ ;          // compute new residue
    $H \leftarrow H + \mathbf{r}$ ;             // accumulate residue for output
    $\mathcal{J} \leftarrow \mathcal{J} \cup \{i_j\}$ ;        // update set of used units
    **if** $\|\mathbf{r}\|_2 < \tau_d$ **then**
        **break** ;            // stop processing when residue is too low
    **else**
        $y \leftarrow f_{i_j}(y)$ ;        // compute feature map
        $\mathbf{h} \leftarrow \frac{\phi_{i_j}(\frac{y}{\|y\|_\infty})}{\left\|\phi_{i_j}(\frac{y}{\|y\|_\infty})\right\|_2}$ ;        // new hash vector using flattened feature map
    **end**
**end**

---

for routing. In this context, similarity is measured by the inner product of flattened feature maps, projected onto different orthogonal subspaces (each unit basis span). Another consequence is that unit weights become specialized in processing a certain type of features, rather than having to adapt to task specific features. This provides the kind of stability needed for continual learning.

For a given unit $\mathcal{U}_k$, $rank(\mathbf{B}_k) \leq m << s$. Hence, it is reasonable to consider that the orthogonal subspace's contribution to total variance is much more important than that of $\mathbf{B}_k$. This is why $\mathcal{H}(x)$ only contains projection residues. Note that in Eq.3 , $\mathbf{h}_{i_k} \in Im(\phi_{i_k})$ and $\|\mathbf{h}_{i_k}\|_2 = 1$. The operand under $\phi_{i_k}$ has an infinite norm of 1, which under Eq.2 leads to inner product bounds independent of input data when considering orthogonality.

Moreover, due to the approximate orthogonality of different feature hashing functions, summing the residues will not lead to much information loss as each residue vector $\mathbf{r}_{i_k}$ is in $Im(\phi_{i_{k-1}})$ but this also explains why each unit can only be selected once. The residues' $\ell_1$-norms are added to the loss function to induce sparsity. Denoting $\mathcal{L}_T$ the specific loss for task $T$ (e.g. KL-divergence for supervised classification), the final loss $\mathcal{L}$ is:

$$\mathcal{L} = \mathcal{L}_T + \lambda \sum_{j \in \{i_0, ..., i_{d-1}\}} \|\mathbf{r}_j\|_1 \tag{6}$$

### 3.3 ONLINE BASIS EXPANSION AND UPDATE

The following paragraphs explain a unit's evolution during training. The described algorithms run each time a unit is selected in Algorithm 1, requiring no external action.

#### 3.3.1 INITIALIZATION AND EXPANSION

Units projection bases are at the heart of the hash-routing algorithm. As explained in section 3.2.1, bases are initially empty and undergo expansion during training. A hash vector (Eq.3) is used to select a unit according to Eq.4. When all units are still empty, a unit is picked randomly and its basis is initialized using the hash vector. Let $\mathcal{I}$ denote the subset of initialized units. When $\mathcal{I} \neq \emptyset$ but some units are still empty, units are still selected according to Eq.4 under the condition that the projection's magnitude is above a minimal threshold $\tau_{empty}$. When $\tau_{empty}$ is not surpassed, a random unit from the remaining empty units is selected instead.

Assuming a unit has been selected as the best for a given hash vector, its basis can expand when the projection's magnitude is below the expansion threshold $\tau_{expand}$. The normalized projection's residue is used as the next basis element. This follows a Gram-Schmidt orthonormalising process to maintain orthonormal basis for each unit. Each basis has a maximum size of $m$ beyond which it cannot expand. The unit selection and expansion algorithms are summarized in Appendix.A.

### 3.3.2 UPDATE

Once a unit basis is full (i.e. it does not contain any zero vector), it still needs to evolve to accommodate routing needs. As the network trains, hashed features will also change and routing might need adjustment. If nothing is done to update full basis, the network might get "stuck" in a bad configuration. Network weights would then need to change in order to compensate for improper routing, resulting in a decrease in performance. Nevertheless, bases should not be updated too frequently as this would lead to instability and units would then need to learn to deal with too many routing configurations.

An *aging* mechanism can be used to stabilize basis update as training progresses. Each time a unit is selected, a counter is incremented and when it reaches its maximum *age*, it is updated. The maximum age can then be increased by means of a geometric progression.

Using the aging mechanism, it becomes possible to apply the update process to bases that are not yet full, thus adding more flexibility. Hence, some bases can expand to include new vectors and update existing ones.

Bases can be updated by replacing vectors that lead to routing instability. Each non-zero basis vector $\mathbf{v}_k$ has a *low projection counter* $c_k$. During training, when a unit has been selected, the basis vector with the lowest projection magnitude sees its low projection counter incremented. The update algorithm is summarized in Algorithm 2.

---

**Algorithm 2:** Unit update

---

**Input:** Current basis (excluding zero-vectors): $\mathbf{B} = (\mathbf{v}_1, ..., \mathbf{v}_m)$,
Current low projection counters: $(c_1, ..., c_m)$,
Current age: $a$, Current maximum age: $\alpha$, Aging rate: $\rho > 1$,
Latest hash vector $\mathbf{h}$
**Output:** Updated basis
**if** $a = \alpha$ **then**
  $\quad i = \arg\max\{c_j\};$         // find basis vector to replace
  $\quad \mathbf{v}_i \leftarrow \mathbf{h} - \mathbf{B}_{-i}\mathbf{h};$     // remove projection on the reduced basis $\mathbf{B}_{-i}$
  $\qquad\qquad\qquad\qquad\qquad$ (without $\mathbf{v}_i$)
  $\quad \mathbf{v}_i \leftarrow \frac{\mathbf{v}_i}{\|\mathbf{v}_i\|_2}$
  $\quad \alpha \leftarrow \rho\alpha;$         // update maximum age
  $\quad a \leftarrow 0;$         // reset age counter
  $\quad c_i \leftarrow 0;$         // reset low projection counter
**else**
  $\quad a \leftarrow a + 1;$         // increment current age
  $\quad i = \arg\min \left\|\mathbf{v}_j^T\mathbf{h}\right\|_2;$     // find low projection counter to increment
  $\quad c_i \leftarrow c_i + 1$

---

### 3.4 TRAINING AND SCALABILITY

HRNs generate feature vectors that can be used for a variety of learning tasks. Given a learning task, optimal network weights can be computed via gradient descent. Feature vectors can be used as input to a fully connected network, to match a given label distribution in the case of supervised learning. As explained in Algorithm 1, each input sample is processed differently and can lead to a different computation graph. Batching is still possible and weight updates only apply to units involved in processing batch data. Weight updates is regularized using the residue vector's norm at each unit level. Low magnitude residue vectors have little contribution to the network's output thus their impact on training of downstream units should be limited. Denoting $\mathcal{L}$ a learning task loss function, $\mathbf{r}$ the hash vector projection residue over a unit's basis, $\mathbf{w}$ the vector of the unit's trainable weights and $\gamma$ a learning rate, regularized weight update of $\mathbf{w}$ becomes:

$$\mathbf{w} \leftarrow \mathbf{w} - \gamma \min(1, \|\mathbf{r}\|_2)\nabla\mathcal{L}(\mathbf{w}) \qquad (7)$$

An HRN can scale simply by adding extra units. Note that adding units between each learning task is not always necessary to insure optimal performance. In our experiments, units were manually added after some learning tasks but this expansion process could be made automatic. Indeed, one or more extra unit(s) could be automatically added whenever all bases have been completely filled. Its architecture could be a copy of an existing unit (chosen randomly).

## 4 RELATED WORK

**Dynamic networks**   Using handcrafted rigid models has obvious limits in terms of scalability. Tanno et al. (2018) builds a binary tree CNN with a routed dataflow. Routing heuristics requires intermediate evaluation on training data. It uses fully connected layers to select a branch. Spring & Shrivastava (2017) builds LSH (Gionis et al., 1999) hash tables of fully connected layer weights to select relevant activations but this does not apply to CNN. Rosenbaum et al. (2017)'s algorithm is closer to our setup. Each sample is processed by different blocks until a maximum processing depth is reached. It uses reinforcement learning to train a router that selects the best processing block at each level, based on a supervised classification scheme. However, their network is task-aware and blocks at each level cannot be used at other levels (unlike HRN units).

**Continual learning**   Parisi et al. (2019) offers a thorough review of state-of-the-art continual learning techniques and algorithms, insisting on a key trade-off: stability vs plasticity. Lomonaco & Maltoni (2017) groups continual learning algorithms into 3 categories: *regularization*, *architectural* and *rehearsal*. Kirkpatrick et al. (2017) introduces a regularization technique using the Fisher information matrix to avoid updating important network weights. Zenke et al. (2017) achieves the same goal by measuring weight importance through its contribution to overall loss evolution across a given number of updates. Rannen et al. (2017) is closer to our setup. The authors continuously train an encoder with different decoders for each task while keeping a stable feature map. Knowledge distillation (Hinton et al., 2015) is used to avoid significant changes to the generated features between each task. A key limitation of this technique is, as mentioned in Rannen et al. (2017), that the encoder will never evolve beyond its inherent capacity as its architecture is frozen. Serra et al. (2018) learns attention nearly-binary masks to avoid updating parts of the network when training for a new task. Similarly, Beaulieu et al. (2020) uses a primary model to modulate the update and response of a secondary model. In both cases, scalability is again limited by the chosen architecture. Li & Hoiem (2017) also uses knowledge distillation in a supervised learning setup but systematically enlarges the last layers to handle new classes. Yoon et al. (2017) limits network expansion by enforcing sparsity when training with extra neurons. Useless neurons are then removed. Xu & Zhu (2018) uses reinforcement learning to optimize network expansion but does not fully take advantage of the inherent network capacity as network weights are frozen before each new task.
Lopez-Paz & Ranzato (2017); Rebuffi et al. (2017); Hayes et al. (2019) store data from previous tasks in various ways to be reused during the current task (rehearsal). Shin et al. (2017); van de Ven & Tolias (2018); Rios & Itti (2018) make use of generative networks to regenerate data from previous tasks. Kamra et al. (2017); Parisi et al. (2018); Kemker & Kanan (2017) use neuroscience inspired concepts such as short-term/long-term memories and a *fear* mechanism to selectively store data during learning tasks, whereas we store a limited number of hashed feature maps in each unit basis, updated using an *aging* mechanism.

## 5 EXPERIMENTS

### 5.1 SETUP

We test our approach in scenarios of increasing complexity and using semantically different datasets. Supervised classification scenarios involve a single HRN that is used across all tasks to generate a feature vector that is fed to different classifiers (one classifier per task). Each classifier is trained only during the task at hand, along with the common HRN. Once the HRN has finished training for a given task, test data from previous tasks is re-encoded using the latest version of the HRN. The new feature vectors are fed into the trained (and frozen) classifiers and accuracy for previous tasks is measured once more.
We compare our approach against 3 other algorithms: a vanilla convolutional network (**VC**) for feature generation with a different classifier per task; Elastic Weight Consolidation (**EWC**) (Kirkpatrick et al., 2017), a typical benchmark for continual learning. Elastic weight consolidation is applied only to a feature generator that feeds into a different classifier per task; Encoder Based Lifelong learning (**ELL**) (Rannen et al., 2017), involving a common feature generator with a different classifier per task. For a fair comparison, we used the same number of epochs per task and the same architecture for classifiers and convolutional layers. For VC, EWC and ELL, the convolutional encoder is equivalent to the unit combination in HRN leading to the largest feature map. Feature codes used in ELL

autoencoders (see Rannen et al. (2017) for more detail) have the same size as the hashed-feature vectors in HRN. For all experiments, we show the evolution of accuracy for the first task (T0) after each task training. This is a clear measure of catastrophic forgetting. We also show the overall accuracy after each task training.

The following scenarios were considered (implementation details can be found in Appendix.C):

**Pairwise-MNIST** Each task is a binary classification of handwritten digits: 0/1, 2/3, ...etc, for a total of 5 tasks (5 epochs each). In this case, tasks are semantically comparable. A 4 units HRN with a depth of 3 was used.

**MNIST/Fashion-MNIST** There are two 10-classes supervised classification tasks, first the Fashion-MNIST dataset, then the MNIST dataset. This a 2 tasks scenario with semantically different datasets. A 6 units HRN (depth of 3) was used for the first task and 2 units were added for the second task.

**SVHN/incremental-Cifar100** This is an 11 tasks scenario, where each task is a 10-classes supervised classification. Task 0 (8 epochs) involves the SVHN dataset. Tasks 1 to 10 (15 epochs each), involve 10 classes out of the 100 classes available in the Cifar100 dataset (new classes are introduced incrementally by groups of 10). All datasets are semantically different, especially task 0 and the others. A HRN of 6 units (depth of 3) was used for the SVHN task and 2 extra units were added before the 10 Cifar100 tasks series. For this experiment, we provide the accuracy drop for each task between its first training and the final task. This is a clear measure of catastrophic forgetting. We also provide the top accuracy score for each task. This measures the network's ability to learn new tasks.

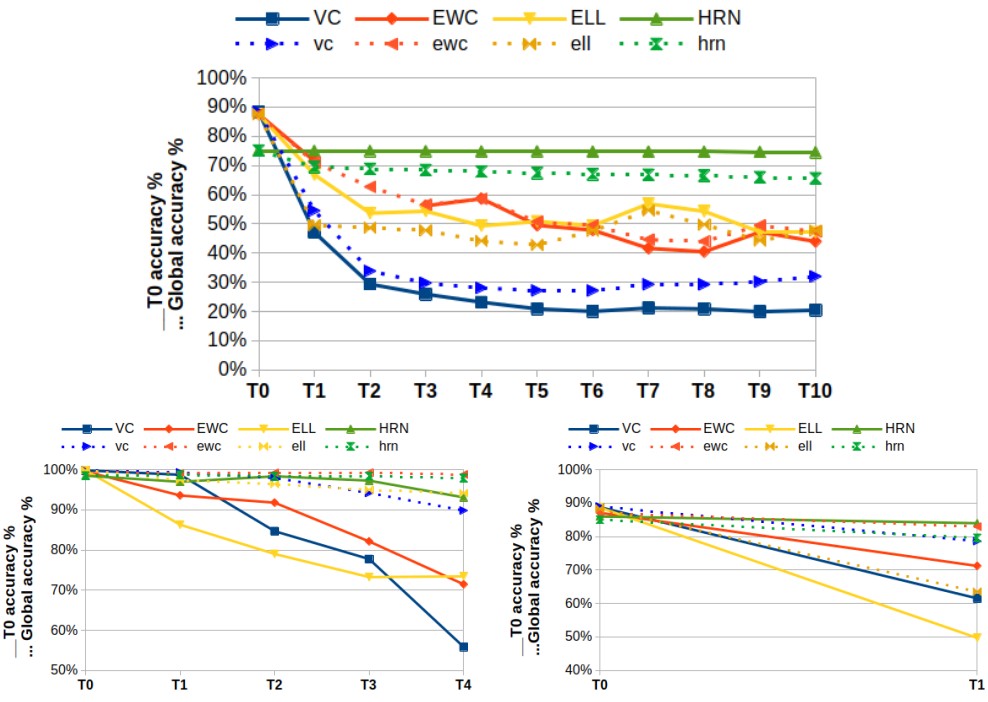

Figure 2: Task 0 accuracy evaluation after each task (continous lines) and global accuracy score (dotted lines). *Top:* SVHN/incremental-Cifar100. *Bottom-left:* Pairwise-MNIST. *Bottom-right:* MNIST/Fashion-MNIST. *Best viewed in color.*

## 5.2 COMPARATIVE ANALYSIS

Figure 2 shows that HRN maintains a stable performance for the initial task, in comparison to other techniques, even in the most complex scenario. Global accuracy plots show that HRN performance degradation for all tasks is very low (even slightly positive in some cases). Moreover, figure 3 shows that catastrophic forgetting is higher with other algorithms, and almost non-existent with HRN. However, 3 also shows that maximum accuracy for each task is often slightly lower in comparison to other algorithms. Indeed, units in a HRN are not systematically updated at each training step and

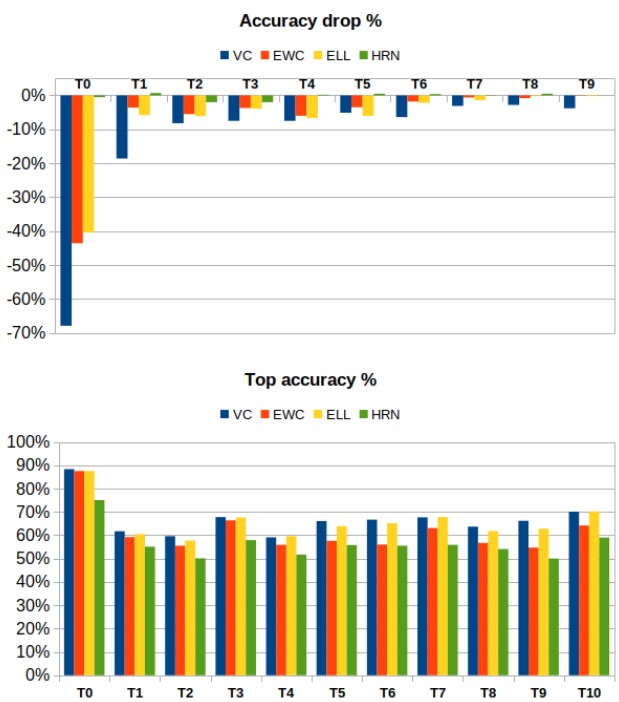

Figure 3: **SVHN/incremental-Cifar100**. *Top:* Final accuracy drop for each task. *Bottom:* Top accuracy score for each task. *Best viewed in color*.

would hence, require a few more epochs to reach top task accuracy as with VC. Furthermore, the trade-off between plasticity and stability can be controlled through the $\lambda$ parameter (see Eq.6) as the ablation study has shown (see B for more detail). Plasticity can also be increased by adding extra units to an HRN.

## 5.3 ROUTING AND NETWORK ANALYSIS

After a few epochs, we observe that some units are used more frequently than others. However, we observe significant changes in usage ratios especially when changing tasks and datasets (see Figure 4). This clearly shows the network's adaptability when dealing with new data. Moreover, some units are almost never used (e.g. $\mathcal{U}_6$ in Figure 4). This shows that the HRN only uses what it needs and that adding extra units does not necessarily lead to better performance.

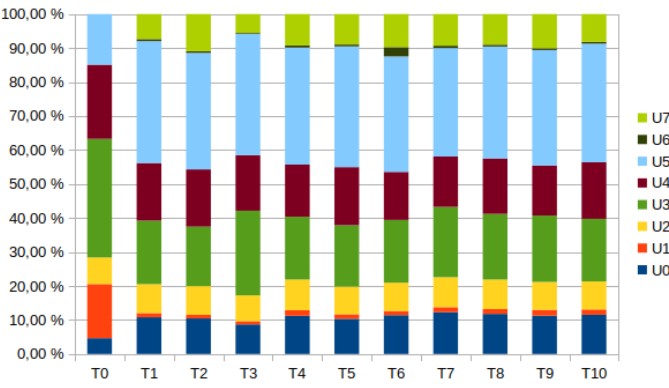

Figure 4: HRN units relative usage ratios for the SVHN/incremental-Cifar100 scenario. Units 6 and 7 were added after T0 (SVHN). *Best viewed in color*.

### 5.4 HYPERPARAMETERS AND ABLATION

Appendix B details the impact of key hyperparameters. Most importantly, the $\ell_1$-norm constraint over the residue vectors (see Eq.6) plays a crucial role in keeping long-term performance but slightly reduces short-term performance. This can be compensated by increasing the number of epochs per task. We have also considered keeping the projection vectors in the output of HRN (the output would be the sum of projection vectors concatenated with the sum of residue vectors) but we saw no significant impact on performance.

## 6 CONCLUSION AND FUTURE WORK

We have introduced the use of feature hashing to generate dynamic configurations in modular convolutional neural networks. Hash-routed convolutional networks generate stable features that exhibit excellent stability and plasticity across a variety of semantically different datasets. Results show excellent feature generation stability, surpassing typical and comparable continual learning benchmarks. Continual supervised learning using HRN still involves the use of different classifiers, even though compression techniques (such as (Chen et al., 2015)) can reduce required memory. This limitation is also a design choice, as it does not limit the use of HRN to supervised classification. Future work will explore the use of HRN in unsupervised and reinforcement learning setups.

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

# A  APPENDIX

## A  SELECTION AND EXPANSION ALGORITHMS

---

**Algorithm 3:** Unit selection and initialization

---

**Input:** Hash vector $\mathbf{h}$,
Initialized units subset $\mathcal{I}$ ($\mathcal{I}^{\complement}$ is the subset of empty units),
Used units subset $\mathcal{J}$
**Output:** Selected unit $\mathcal{U}_i$
**if** $\mathcal{I} = \emptyset$ **then**
    Select a random unit $\mathcal{U}_i$
    $\mathbf{B}_i \leftarrow (\mathbf{h})$ ;         // initialize its basis using $\mathbf{h}$
**else**
    **if** $\mathcal{I}^{\complement} = \emptyset$ **or** $\max_{j \in \mathcal{I} \setminus \mathcal{J}} \|\mathbf{B_j h}\|_2 \geq \tau_{empty}$ **then**
        Select unit according to Eq.4
    **else**
        Select a random unit $\mathcal{U}_i$ from the remaining empty units
        $\mathbf{B}_i \leftarrow (\mathbf{h})$ ;         // initialize its basis using $\mathbf{h}$

---

**Algorithm 4:** Unit expansion

---

**Input:** Hash vector $\mathbf{h}$, initialized unit $\mathcal{U}_i$
$\mathbf{p} = \mathbf{B}_i \mathbf{h}$
$\mathbf{r} = \mathbf{h} - \mathbf{p}$
**if** $\|\mathbf{p}\|_2 < \tau_{expand}$ **and** $nonzero(\mathbf{B}_i) < m$ **then**
    $\mathbf{B}_i \leftarrow \left(\mathbf{B}_i, \frac{\mathbf{r}}{\|\mathbf{r}\|}\right)$ ;        replace a zero vector with the normalized residue

---

## B   HYPERPARAMETERS AND ABLATION EXPERIMENTS

| Experiment | Hyperparameters | Metrics |
|---|---|---|
| Baseline | $gradUpdate = True$
$m = 3$
$s = 100$
$\tau_{expand} = 0.01$
$\rho = 1.2$
$\lambda = 1.0$ | T0 accuracy = 96.41%
Max task accuracy = 98.87%
Global accuracy = 95.24%
Training time = 59.3 min |
| Gradient update | $gradUpdate = False$ | T0 accuracy = 97.3%
Max task accuracy = 98.87%
Global accuracy = 92.55%
Training time = 55 min |
| Low residue constraint | $\lambda = 0$ | T0 accuracy = 96.41%
Max task accuracy = 99.62%
Global accuracy = 95.24%
Training time = 59.7 min |
| High residue constraint | $\lambda = 10$ | T0 accuracy = 99.24%
Max task accuracy = 74.94%
Global accuracy = 92.58%
Training time = 59.8 min |
| Low aging rate | $\rho = 1.2$ | T0 accuracy = 99.05%
Max task accuracy = 97.73%
Global accuracy = 93.47%
Training time = 58 min |
| High aging rate | $\rho = 10$ | T0 accuracy = 97.45%
Max task accuracy = 99.20%
Global accuracy = 89.77%
Training time = 58.8 min |
| Low basis size | $m = 2$ | T0 accuracy = 98.25%
Max task accuracy = 92.81%
Global accuracy = 94.11%
Training time = 56.8 min |
| High basis size | $m = 5$ | T0 accuracy = 99.10%
Max task accuracy = 83.03%
Global accuracy = 92.44%
Training time = 60 min |
| Low embedding size | $s = 50$ | T0 accuracy = 96.83%
Max task accuracy = 86.62%
Global accuracy = 87.26%
Training time = 58 min |
| High embedding size | $s = 500$ | T0 accuracy = 99.67%
Max task accuracy = 99.67%
Global accuracy = 98.50%
Training time = 58.1 min |
| Low expansion threshold | $\tau_{expand} = 0.001$ | T0 accuracy = 98.06%
Max task accuracy = 98.72%
Global accuracy = 96.33%
Training time = 58.6 min |
| High expansion threshold | $\tau_{expand} = 0.1$ | T0 accuracy = 97.45%
Max task accuracy = 98.63%
Global accuracy = 93.15%
Training time = 58.6 min |

Table 1: Hyperparameters' ranges and observed impact (averaged over 10 runs) over accuracy and training/inference time, based on the Pairwise-MNIST scenario.

## C  IMPLEMENTATION DETAILS

This section details the network architectures and hyperparameters that were used for each experiment.

| Layer | Type | Parameters |
|---|---|---|
| 1 | Conv 2D | Filters: 16, Kernel: $5 \times 5$
Stride: $3 \times 3$, Padding: $2 \times 2$
BatchNorm2D, Activation: LeakyReLU |
| 2 | Conv 2D | Filters: 32, Kernel: $2 \times 2$
Stride: $3 \times 3$, Padding: 0
BatchNorm2D, Activation: LeakyReLU |
| 3 | Dense | Neurons: 60, Activation: LeakyReLU |
| 4 | DropOut | dropProb: 0.2 |
| 5 | Dense | Neurons: 10, Activation: LeakyReLU |
| 6 | DropOut | dropProb: 0.2 |
| 7 | Dense | Neurons: 10, Activation: None |

| Hyperparameter | Value |
|---|---|
| batch size | 16 |
| epochs | 5,5,5,5,5 |
| Optimizer | Adam |
| learning rate | 0.001 |

Table 2: **VC:** Pairwise-MNIST network architecture and hyperparameters

| Layer | Type | Parameters |
|---|---|---|
| 1 | Conv 2D | Filters: 16, Kernel: $5 \times 5$
Stride: $3 \times 3$, Padding: $2 \times 2$
BatchNorm2D, Activation: LeakyReLU |
| 2 | Conv 2D | Filters: 32, Kernel: $2 \times 2$
Stride: $3 \times 3$, Padding: 0
BatchNorm2D, Activation: LeakyReLU |
| 3 | Dense | Neurons: 60, Activation: LeakyReLU |
| 4 | DropOut | dropProb: 0.2 |
| 5 | Dense | Neurons: 10, Activation: LeakyReLU |
| 6 | DropOut | dropProb: 0.2 |
| 7 | Dense | Neurons: 10, Activation: None |

| Hyperparameter | Value |
|---|---|
| batch size | 16 |
| epochs | 5,5,5,5,5 |
| Optimizer | Adam |
| learning rate | 0.001 |
| Fisher matrix samples | 200 |
| $\lambda$ | 1024 |

Table 3: **EWC:** Pairwise-MNIST network architecture and hyperparameters

| Layer | Type | Parameters |
|---|---|---|
| 1 | Conv 2D | Filters: 16, Kernel: $5 \times 5$
Stride: $3 \times 3$, Padding: $2 \times 2$
BatchNorm2D, Activation: LeakyReLU |
| 2 | Conv 2D | Filters: 32, Kernel: $2 \times 2$
Stride: $3 \times 3$, Padding: 0
BatchNorm2D, Activation: LeakyReLU |
| 3 | Dense | Neurons: 60, Activation: LeakyReLU |
| 4 | DropOut | dropProb: 0.2 |
| 5 | Dense | Neurons: 10, Activation: LeakyReLU |
| 6 | DropOut | dropProb: 0.2 |
| 7 | Dense | Neurons: 10, Activation: None |

| Hyperparameter | Value |
|---|---|
| batch size | 16 |
| epochs | 5,5,5,5,5 |
| Optimizer | Adam |
| learning rate | 0.001 |
| Codes length | 100 |
| Embedding Size | 288 |
| Temperature | 3.0 |
| Stabilization epochs | 2 |
| Stabilization learning rate | 0.001 |

Table 4: **ELL:** Pairwise-MNIST network architecture and hyperparameters

| Units Quantity | Unit Layers | Parameters |
|---|---|---|
| 2 | Conv 2D | Filters: 16, Kernel: $5 \times 5$
Stride: $3 \times 3$, Padding: $2 \times 2$
BatchNorm2D
Activation: LeakyReLU |
| 2 | Conv 2D | Filters: 16, Kernel: $2 \times 2$
Stride: $3 \times 3$, Padding: 0
BatchNorm2D
Activation: LeakyReLU |

| Classifier Layer | Type | Parameters |
|---|---|---|
| 1 | Dense | Neurons: 60
Activation: LeakyReLU |
| 2 | DropOut | dropProb: 0.2 |
| 3 | Dense | Neurons: 10
Activation: LeakyReLU |
| 4 | DropOut | dropProb: 0.2 |
| 5 | Dense | Neurons: 10
Activation: None |

| Hyperparameter | Value |
|---|---|
| batch size | 16 |
| epochs | 5,5,5,5,5 |
| extra units | 0,0,0,0,0 |
| Optimizer | Adam |
| learning rate | 0.001 |
| $d$ | 3 |
| $m$ | 3 |
| $s$ | 300 |
| $\tau_d$ | 0.2 |
| $\tau_{expand}$ | 0.01 |
| $\alpha$ | 5 |
| $\rho$ | 1.2 |
| $\lambda$ | 1.0 |

Table 5: **HRN:** Pairwise-MNIST units/classifiers architecture and hyperparameters

| Layer | Type | Parameters |
|---|---|---|
| 1 | Conv 2D | Filters: 32, Kernel: $5 \times 5$
Stride: $3 \times 3$, Padding: $2 \times 2$
BatchNorm2D, Activation: LeakyReLU |
| 2 | Conv 2D | Filters: 64, Kernel: $2 \times 2$
Stride: $3 \times 3$, Padding: 0
BatchNorm2D, Activation: LeakyReLU |
| 3 | Dense | Neurons: 60, Activation: LeakyReLU |
| 4 | DropOut | dropProb: 0.2 |
| 5 | Dense | Neurons: 10, Activation: LeakyReLU |
| 6 | DropOut | dropProb: 0.2 |
| 7 | Dense | Neurons: 10, Activation: None |

| Hyperparameter | Value |
|---|---|
| batch size | 16 |
| epochs | 20,10 |
| Optimizer | Adam |
| learning rate | 0.001 |

Table 6: **VC:** Fashion-MNIST/MNIST network architecture and hyperparameters

| Layer | Type | Parameters |
|---|---|---|
| 1 | Conv 2D | Filters: 32, Kernel: $5 \times 5$
Stride: $3 \times 3$, Padding: $2 \times 2$
BatchNorm2D, Activation: LeakyReLU |
| 2 | Conv 2D | Filters: 64, Kernel: $2 \times 2$
Stride: $3 \times 3$, Padding: 0
BatchNorm2D, Activation: LeakyReLU |
| 3 | Dense | Neurons: 60, Activation: LeakyReLU |
| 4 | DropOut | dropProb: 0.2 |
| 5 | Dense | Neurons: 10, Activation: LeakyReLU |
| 6 | DropOut | dropProb: 0.2 |
| 7 | Dense | Neurons: 10, Activation: None |

| Hyperparameter | Value |
|---|---|
| batch size | 16 |
| epochs | 20,10 |
| Optimizer | Adam |
| learning rate | 0.001 |
| Fisher matrix samples | 200 |
| $\lambda$ | 400 |

Table 7: **EWC:** Fashion-MNIST/MNIST network architecture and hyperparameters

| Layer | Type | Parameters |
|---|---|---|
| 1 | Conv 2D | Filters: 32, Kernel: $5 \times 5$
Stride: $3 \times 3$, Padding: $2 \times 2$
BatchNorm2D, Activation: LeakyReLU |
| 2 | Conv 2D | Filters: 64, Kernel: $2 \times 2$
Stride: $3 \times 3$, Padding: 0
BatchNorm2D, Activation: LeakyReLU |
| 3 | Dense | Neurons: 60, Activation: LeakyReLU |
| 4 | DropOut | dropProb: 0.2 |
| 5 | Dense | Neurons: 10, Activation: LeakyReLU |
| 6 | DropOut | dropProb: 0.2 |
| 7 | Dense | Neurons: 10, Activation: None |

| Hyperparameter | Value |
|---|---|
| batch size | 16 |
| epochs | 20,10 |
| Optimizer | Adam |
| learning rate | 0.001 |
| Codes length | 300 |
| Embedding Size | 576 |
| Temperature | 3.0 |
| Stabilization epochs | 3 |
| Stabilization learning rate | 0.001 |

Table 8: **ELL:** Fashion-MNIST/MNIST network architecture and hyperparameters

| Units Quantity | Unit Layers | Parameters |
|---|---|---|
| 3 | Conv 2D | Filters: 6, Kernel: $5 \times 5$
Stride: $3 \times 3$, Padding: $2 \times 2$
Activation: LeakyReLU |
| 3 | Conv 2D | Filters: 8, Kernel: $2 \times 2$
Stride: $3 \times 3$, Padding: 0
Activation: LeakyReLU |

| Classifier Layer | Type | Parameters |
|---|---|---|
| 1 | Dense | Neurons: 60
Activation: LeakyReLU |
| 2 | DropOut | dropProb: 0.2 |
| 3 | Dense | Neurons: 10
Activation: LeakyReLU |
| 4 | DropOut | dropProb: 0.2 |
| 5 | Dense | Neurons: 10
Activation: None |

| Hyperparameter | Value |
|---|---|
| batch size | 16 |
| epochs | 20,10 |
| extra units | 0,2 |
| Optimizer | Adam |
| learning rate | 0.001 |
| $d$ | 3 |
| $m$ | 3 |
| $s$ | 300 |
| $\tau_d$ | 0.2 |
| $\tau_{expand}$ | 0.01 |
| $\alpha$ | 5 |
| $\rho$ | 10 |
| $\lambda$ | 1.0 |

Table 9: **HRN:** Fashion-MNIST/MNIST network architecture and hyperparameters

| Layer | Type | Parameters |
|-------|------|------------|
| 1 | Conv 2D | Filters: 36, Kernel: $3 \times 3$
Stride: $2 \times 2$, Padding: $2 \times 2$
BatchNorm2D
Activation: LeakyReLU |
| 2 | Conv 2D | Filters: 99, Kernel: $2 \times 2$
Stride: $1 \times 1$, Padding: $1 \times 1$
BatchNorm2D
Activation: LeakyReLU |
| 3 | DropOut 2D | dropProb: 0.5 |
| 4 | Dense | Neurons: 200, BatchNorm
Activation: ReLU |
| 5 | DropOut | dropProb: 0.4 |
| 6 | Dense | Neurons: 100, BatchNorm
Activation: ReLU |
| 7 | DropOut | dropProb: 0.4 |
| 8 | Dense | Neurons: 100, BatchNorm
Activation: None |
| 9 | DropOut | dropProb: 0.2 |

| Hyperparameter | Value |
|----------------|-------|
| batch size | 32 |
| epochs | 8,15,15,15
15,15,15,15
15,15,15 |
| Optimizer | Adam |
| learning rate | 0.001 |

Table 10: **VC:** SVHN/incremental-Cifar100 network architecture and hyperparameters

| Layer | Type | Parameters |
|-------|------|------------|
| 1 | Conv 2D | Filters: 36, Kernel: $3 \times 3$
Stride: $2 \times 2$, Padding: $2 \times 2$
BatchNorm2D
Activation: LeakyReLU |
| 2 | DropOut 2D | dropProb: 0.3 |
| 3 | Conv 2D | Filters: 99, Kernel: $2 \times 2$
Stride: $1 \times 1$, Padding: $1 \times 1$
BatchNorm2D
Activation: LeakyReLU |
| 4 | DropOut 2D | dropProb: 0.3 |
| 5 | Dense | Neurons: 200, BatchNorm
Activation: ReLU |
| 6 | DropOut | dropProb: 0.5 |
| 7 | Dense | Neurons: 100, BatchNorm
Activation: ReLU |
| 8 | DropOut | dropProb: 0.4 |
| 9 | Dense | Neurons: 100, BatchNorm
Activation: None |
| 10 | DropOut | dropProb: 0.2 |

| Hyperparameter | Value |
|----------------|-------|
| batch size | 32 |
| epochs | 8,15,15,15
15,15,15,15
15,15,15 |
| Optimizer | Adam |
| learning rate | 0.001 |
| Fisher matrix samples | 200 |
| $\lambda$ | 250 |

Table 11: **EWC:** SVHN/incremental-Cifar100 network architecture and hyperparameters

| Layer | Type | Parameters |
|-------|------|------------|
| 1 | Conv 2D | Filters: 36, Kernel: $3 \times 3$
Stride: $2 \times 2$, Padding: $2 \times 2$
BatchNorm2D
Activation: LeakyReLU |
| 2 | Conv 2D | Filters: 99, Kernel: $2 \times 2$
Stride: $1 \times 1$, Padding: $1 \times 1$
BatchNorm2D
Activation: LeakyReLU |
| 3 | DropOut 2D | dropProb: 0.5 |
| 4 | Dense | Neurons: 200, BatchNorm
Activation: ReLU |
| 5 | DropOut | dropProb: 0.4 |
| 6 | Dense | Neurons: 100, BatchNorm
Activation: ReLU |
| 7 | DropOut | dropProb: 0.4 |
| 8 | Dense | Neurons: 100, BatchNorm
Activation: None |
| 9 | DropOut | dropProb: 0.2 |

| Hyperparameter | Value |
|----------------|-------|
| batch size | 16 |
| epochs | 8,15,15,15
15,15,15,15
15,15,15 |
| Optimizer | Adam |
| learning rate | 0.001 |
| Codes length | 2800 |
| Embedding Size | 32076 |
| Temperature | 3.0 |
| Stabilization epochs | 3 |
| Stabilization learning rate | 0.001 |

Table 12: **ELL:** SVHN/incremental-Cifar100 network architecture and hyperparameters

| Units Quantity | Unit Layers | Parameters |
|----------------|-------------|------------|
| 2 | Conv 2D | Filters: 36, Kernel: $3 \times 3$
Stride: $2 \times 2$, Padding: $2 \times 2$
Activation: LeakyReLU |
| | DropOut 2D | dropProb: 0.5 |
| 2 | Conv 2D | Filters: 36, Kernel: $3 \times 3$
Stride: $1 \times 1$, Padding: $2 \times 2$
Activation: LeakyReLU |
| | DropOut 2D | dropProb: 0.5 |
| 1 | Conv 2D | Filters: 12, Kernel: $2 \times 2$
Stride: $1 \times 1$, Padding: $1 \times 1$
Activation: ReLU |
| | DropOut 2D | dropProb: 0.5 |
| 1 | Conv 2D | Filters: 24, Kernel: $4 \times 4$
Stride: $2 \times 2$, Padding: $1 \times 1$
Activation: LeakyReLU |
| | DropOut 2D | dropProb: 0.5 |

| Classifier Layer | Type | Parameters |
|------------------|------|------------|
| 1 | Dense | Neurons: 200
Activation: ReLU |
| 2 | DropOut | dropProb: 0.4 |
| 3 | Dense | Neurons: 100
Activation: ReLU |
| 4 | DropOut | dropProb: 0.4 |
| 5 | Dense | Neurons: 100
Activation: None |
| 6 | DropOut | dropProb: 0.2 |

| Hyperparameter | Value |
|----------------|-------|
| batch size | 16 |
| epochs | 8,15,15,15
15,15,15,15
15,15,15 |
| extra units | 0,2,0,0
0,0,0,0
0,0,0 |
| Optimizer | Adam |
| learning rate | 0.001 |
| $d$ | 3 |
| $m$ | 3 |
| $s$ | 2800 |
| $\tau_d$ | $1.10^{-5}$ |
| $\tau_{expand}$ | 0.01 |
| $\alpha$ | 5 |
| $\rho$ | 10 |
| $\lambda$ | 1.0 |

Table 13: **HRN:** SVHN/incremental-Cifar100 network architecture and hyperparameters

## D    RUNTIME PERFORMANCE

This section compares the training and inference run times of each algorithm, considering the SVHN/incremental-Cifar100 experiment. All runs were performed on a 32 CPUs machine with an Nvidia P100 GPU and 16 GB of RAM. Performance figures have been averaged over 10 runs. Batch size is 16 for all runs and algorithms. Note that one training (*resp.* test) epoch for SVHN corresponds to 73257 (*resp.* 26032) samples; 1 training (*resp.* test) epoch for incremental-Cifar100 corresponds to 5000 (*resp.* 1000) samples. HRN typically require more time for training and in-

|  |  | **SVNH** 
 1 epoch | **Incremental-Cifar100** 
 1 epoch |
|---|---|---|---|
| **VC** | Training | 29s | 3s |
|  | Inference | 6s | 1s |
| **EWC** | Training | 29s | 3s |
|  | Inference | 6s | 1s |
| **ELL** | Training+ 
 (4 epochs 
 Stabilization) | 39s 
 (9min) | 34s 
 (36s) |
|  | Inference | 6s | 1s |
| **HRN** | Training | 11min14s | 46s |
|  | Inference | 2min24s | 6s |

Table 14: Training and inference run times (10 runs average) for each algorithm.

ference compared to a classical convolutional neural network. This is due to dynamic routing, as it is hardly compatible with batch processing and therefore exhibits poor performance on GPU. Each batch is split into individual samples and each sample is processed differently in the dynamic network. A more adequate processing backend would be a Many-cores/Network-On-Chip processor or an FPGA.

