# OpenReview forum: "Continual learning using hash-routed convolutional neural networks"
_ICLR.cc/2021/Conference — Reject_

### Official Review · AnonReviewer1 · 2020-10-28
**The paper has values in the exploration of data hashing for neural networks. At the same time it needs more evidence make sense the motivation.**

**Rating:** 6
**Confidence:** 4

**Review:**

##########################################################################

Summary:

This paper studies the problem of continual learning and proposes a new learning framework named hash-routed convolutional neural networks (HRN). HRN has a set of convolutional units and hashes similar data to the same unit. With this design, the paper claims three key contributions. (1) HRN provides excellent plasticity and more stable features. (2) HRN achieves excellent performance on a variety of benchmarks. (3) HRN can be used for unsupervised or reinforcement learning.


##########################################################################

Reasons for score:

Overall, I like the scope of this paper that studies continual learning. The experiments in Figure 2 verify that the proposed HRN outperforms a number of baselines on incremental-Cifar100 dataset. However, I am not sure whether the dataset makes sense for benchmarking continual learning. It seems the authors split the 100 classes into 10 groups, with each group having 10 classes. The 10 groups and corresponding labels serve as 10 distinct tasks for evaluating continual learning. Such formulation is odd that leads to decreasing accuracy scores as shown in Figure 2. I cannot find a real-world application that can benefit from such formulation due to decreased accuracy scores. It will be good if the paper can clarity the above points to strengthen its motivation.

##########################################################################

Pros:

1. The paper proposes a novel neural network HRN for continual learning. HRN leverages multiple units of CNN and hashes similar data to the same unit for training.

2. The proposed HRN achieves impressive performance when compared with a selected set of baselines. HRN consistently shows more robust accuracy scores on three datasets, as shown in Figure 2 and Table 1.


##########################################################################

Cons:

The most important point relates to the motivation of problem formulation. The paper splits a 100-classes dataset into 10 10-classes datasets. Any two datasets have no overlapping in the classes. Under such formulation, a model suffers from decreased accuracy score as shown in Figure 2. It is necessary to explain why decreased scores are appealing in practice given they are the base of the concerned continual learning.

According to Figure 2 and Table 1, all models suffer from decreased accuracy scores. Besides, the accuracy scores decrease monotonously regarding the task ids. Task id should not be a factor of performance. It will be good if the paper can provide some explanations. All existing machine learning researchers seek better accuracy scores so the decreased accuracy scores seem unusual.

It may not be fair to compare baselines with HRN that exhibits a larger model size. HRN leverages multiple (e.g. 6 in figure 2) units of CNN. To be fair, the paper should use the same number of units (or equivalently ensembles) for a baseline method. It will be good if the paper can run some experiments or show a comparison on model sizes.


##########################################################################

Questions during rebuttal:

It will be nice if the authors can add experiments or discussions related to decreased accuracy scores. It will be perfect if the authors can show real applications of the proposed continual learning with decreased accuracy scores.

---

> ### Author Response · Authors · 2020-11-19
> **R1 specific comments**
>
> - Incremental-Cifar100 dataset construction: R1 is correct in saying that splitting a 100 classes dataset into 10 10 classes dataset has no practical application. This is merely an "easy" way to generate distinct learning tasks in order to assess catasptrophic forgetting and learning capacity for a continual learning model. The same idea has been applied for the Pairwise-MNIST dataset. This kind of trick has indeed no real application but is very commonly used in continual learning benchmarks, as there are very few datasets that were specifically developped to test continual learning models.
> We have explored more realistic uses of continual learning when we combined learning tasks from MNIST and Fashion-MNIST, and SVHN/Cifar100.
> - Fairness w.r.t model sizes: It is true that for each experiment, HRN contains more trainable parameters and convolution layers than other methods. However, adding extra convolution layers to other methods would also not be fair, as this would increase the feature map size before classification. Nevertheless, HRN have a limited depth and at each forward pass, only a given combination of units is involved, resulting in a limited feature map size (even before feature hashing).

---

### Official Review · AnonReviewer3 · 2020-10-28
**Hashing is an interesting idea for routing networks, but I would like to see a more complete evaluation and a clearer description of the method**

**Rating:** 4
**Confidence:** 4

**Review:**

############## Summary ##############

This submission addresses the question of how to avoid catastrophic forgetting while being scalable and adaptable to multiple tasks. The paper introduces the hash-routed network (HRN), a new routing mechanism for neural networks whereby feature hashing (Weinberger et al.) is used to determine the similarity between data points; similar data points are then routed to the same units. HRNs  work by keeping various convolutional units, each coupled with an orthonormal projection basis. The parameters of the convolutional layers are trained via vanilla back-propagation and the orthonormal basis is created following the Gram Schmidt procedure. An aging mechanism enables updating basis vectors that have become obsolete over time. The proposed method is evaluated in three standard continual learning settings, and is demonstrated to avoid catastrophic forgetting.

############## Strengths ##############

1. The idea to use hashing to determine routing paths is novel and could be interesting to the general deep learning community.
2. This is to my knowledge the first mechanism that enables learning routing networks for lifelong learning, which is certainly relevant.
3. Code is included for reproducibility.

############## Weaknesses ##############

1. The obtained results show that, while catastrophic forgetting is avoided, this comes at the cost of a model that is too inflexible to handle new tasks.
2. The proposed algorithm hinges too heavily on heuristics, including counters, thresholds, and manually deciding when new units should be added. While this is not necessarily wrong, it would require substantial experimentation and validation that each component is necessary to achieving the desired behavior.
3. The paper is hard to follow and so it makes it tough to assess exactly how the different parts of the proposed method are implemented.

############## Recommendation ##############

Unfortunately, at this time I have to recommend this paper for rejection. I think a future iteration of this work could be interesting to the community if either much of the heuristics were replaced by more well-founded techniques, or each of the heuristics was appropriately analyzed in terms of its empirical effect. Moreover, the paper should provide other views of the obtained results, including final overall accuracy, which is omitted.

############## Arguments ##############

The key idea studied in this work is how to leverage hashing to decide different routes in a routing net in a continual learning setting. I believe this to be a powerful and interesting idea, but I am of the opinion that the proposed approach uses too many heuristic tricks, and therefore makes it hard to assess the benefits of the key contribution. In particular, the method uses a threshold (1) on the projection residual magnitude to decide when to stop chaining, a threshold (2) on the projection magnitude to decide when to pick a random, un-initialized unit, another threshold (3) on the projection magnitude to decide when to add a new basis vector to the chosen unit, and a final threshold (4) on the age of a basis to choose when to update it. While it is possible that such a heuristic method is the right way to go, it warrants a much more comprehensive evaluation that enables us to understand what the contribution of each of the aspects is to overall performance.

On the other hand, it seems that the obtained results are not very strong, and so I didn't find enough evidence in the paper to convince me that the proposed method is appropriate for continual learning. In particular, it seems that the cost of avoiding forgetting is to almost entirely prevent the model from adapting to new tasks (e.g., Table 1-top). It would be much easier to evaluate these results if the authors provided other views into their results. For example, the authors could show accuracy evaluation of all tasks throughout the training process, average final accuracy of all tasks, average forgetting across all tasks.

The other major point I have is the lack of clarity in the paper. Section 2 is supposed to provide background for feature hashing. Ideally, the reader should finish reading this section and come out with an understanding of what the point of hashing is in general, how it's computed, and how it will be useful for the provided approach. Instead, this section gives a few mathematical properties of feature hashing, with no intuition or high-level description of it or how it will be used. It seems like the rest of the paper similarly doesn't really give an intuition for why hashing might be useful, other than the fact that it gives a constant-size dimensionality to the hashed features.  The manuscript places emphasis on the fact that inner product is maintained under hashing, so I suppose the point is that distance metrics can be computed in the hashed space, which is used to compute similarity. I believe this fact is never explicitly stated, even though it is a key point of the proposed approach.

I'm also surprised that the paper did not provide a comparison to stronger lifelong baselines, like those based on experience replay, which have been shown to vastly outperform regularization-based methods in recent work (e.g., Lopez-Paz & Ranzato).

############## Additional feedback ##############

The following points are provided as feedback to hopefully help better shape the submitted manuscript, but did not impact my recommendation in a major way.

Abstract
- The abstract mentions that the method is potentially useful for unsupervised or reinforcement learning unlike prior approaches, but this is never tested. Why should the reader believe this to be the case?

Intro
- Some paragraphs are spaced and some aren't. The rest of the paper has un-spaced paragraphs only. Please be sure to use consistent formatting.
- What are "stable features", and how are they useful for unsupervised learning? The manuscript states that the proposed approach would find such stable features, but I didn't see any evidence of this in the evaluation.
- The paper initially says that the method will be trained only with gradient descent, but then it mentions regularization techniques. Would it be fair to say that EWC is also trained only with gradient descent, since it only adds a regularization term and then uses the penalized objective for gradient descent?
- If the three benchmarks are MNIST, Fashion/MNIST, and SVHN/incremental-Cifar100, then the text shouldn't say that benchmarks "include" those three, but rather that those three "are" the benchmarks used.

Sec 2
- I recommend using `` '' for quotation marks
- This section focuses on providing some theoretical properties of feature hashing, but no good intuition about what it's doing, how it's computed, or how it will be useful for the approach.
    - What's N in the summation? In Weinberger et al., there's no N. It seems to be j=1^N where h(j)=i.
    - Need to define \sigma.

Sec 3.1
- Elements of the proposed architecture are introduced, but we're still given no intuition of what they'll be useful for. This section should guide the reader by explaining from the beginning what the purpose of each piece of the architecture will be.

Sec 3.2.1
- Why are feature maps of vanilla CNNs growing in size?
- What is the i index? It seems to be the selected unit. Oddly, in Sec 3.1 k was the selected unit, but here it seems that k is the operation number in the chain of operations.
- Does the set subtraction in Eq. 4 mean that the units used so far cannot be used for the subsequent operations? This should be stated in text.
- Why is there no mention of how the bases are initialized yet?

Sec 3.2.2
- Where is there an inner product between hashes computed for similarity? Is this the inner product with the basis? We can't know that at this point, since we have no information about how the bases are created.
- Overall, this section could clarify a lot of things like why is orthogonality important in this context, why the residuals are chosen. It attempts to do so, but leaves me somewhat confused still.
    - This statement: "the orthogonal subspace’s contribution to total variance is much more important than that of B_k" could be explained in more detail. Does this mean that, since hashes are similar, there is not much variance across projections and so the residual contains more information?
- Why are residuals encouraged to be sparse?

Sec 3.3.1
- basis --> bases (plurals)
- Using two separate thresholds on the projection magnitude seems to introduce too many hyperparameters. It seems odd to use projection magnitude simultaneously for 1) choosing the current unit, 2) choosing whether to initialize a new unit, and 3) choosing when to expand a unit's basis projection. Why do this?

Sec 3.4
- Scalability depends on manually adding units. Would it be possible to come up with an automatic way to do this? This section states that it should be possible, but I think a significant contribution of this work could come from designing a technique for it.
- This manual addition seems to go against the stated goal of keeping data scientist involvement to a minimum.

Sec 5
- Is EWC also allowed a task-specific classifier? A fair evaluation would give EWC a task-specific classifier like other methods get.
- The description of the experimental setup is fairly complete, including details about when units were added, and which data sets were "semantically different"
- Table 1 uses "," instead of "." for decimals. The "%" sign is missing.
- The fact that catastrophic forgetting is avoided at the cost of lower overall performance suggests that the stability-plasticity trade-off was not well chosen: there is too much stability at the cost of very little flexibility.
- We don't get to see average performance to accurately assess performance in Table 1.
    - Manually computing it for SVHN/CIFAR-100 gives 55.46 (HRN) - 58.93 (ELL).
- How were parameters of baselines chosen? It's odd that EWC neither retains performance on earlier tasks nor adapts to new tasks. This suggests a very poor choice of \lambda.
- It's nice to see unit usage plots, but these don't really show substantial differences in usage across tasks: all bars except the first look about the same.
- The hyperparameters and ablation evaluation does not seem to add much value. Here, I would like to see a much more comprehensive ablation study that goes in depth into the effect of the different parts of the proposed algorithm.

---

> ### Author Response · Authors · 2020-11-19
> **R3 specific comments**
>
> - A more detailed ablation study was added to the Appendix, showing the impact that each threshold has on accuracy, catastrophic forgetting and runtime. Typical trade-offs are performance vs runtime and top accuracy vs catastrophic forgetting mitigation.
> - We have added plots for global accuracy, accuracy drop and top task accuracy, removing the need for Table 1. R3 is correct in saying that better catastrophic forgetting mitigation comes at the cost of lower per-task accuracy. However, this can be compensated by increasing training epochs or by adding extra units.
> - Sections 2 and 3.1 have been modified to better explain the intution behind the choice of feature hashing and its relevance to our work. Also note that section 3.2.2 already discusses key design choices.
> - As explained to R2, we do not claim to surpass state-of-the-art continual learning algorithms with HRN. We merely present a novel and scalable techique. This is why we have compared our approach only with the most relevant approaches and typical benchmarks.
> - Stable features: HRN's output is a feature vector. We have coupled it to MLP classifiers for supervised tasks but we could have used the generated feature vector to estimate a V-value and a policy, in a policy gradient reinforcement learning setup. We also could have reconstructed the input using the generated feature vector, through the use of an upsampling transpose-convolution block, in an unsupervised learning setup. This is why HRN generate stable features that could be used for unsupervised or reinforcement learning.
> - Section 3.2.2: R3 says: "Does this mean that, since hashes are similar, there is not much variance across projections and so the residual contains more information?" This is the correct interpretation.
> - Empirically, as shown in the ablation study, residuals sparsity increases catastrophic forgetting mitigation.
> - Section 3.4 now explains how to automate the addition of new units.

---

> > ### Comment · AnonReviewer3 · 2020-11-20
> > **Improved clarity, but heuristics are still unmotivated, results sub-par, and baselines insufficient**
> >
> > Thank you for providing a response to a lot of my comments, and for updating the manuscript to address some of my concerns. Please find my answers to your comments below.
> >
> > - The included ablation study is not analyzed in much detail.
> >     - The authors tried a few values for some of the  hyper-parameters and included a table of results, but didn't discuss the findings at all.
> >     - What should I get from that table? How can I choose these values myself?
> >     - The biggest impact seems to come from using a larger embedding size and a lower expansion threshold, but these are not discussed.
> >     - The low aging rate is exactly the same as the "baseline" aging rate, but results are different.
> >     - Why is this test run in the simplest benchmark (pairwise MNIST)?
> >     - It still doesn't give any intuition as to why each heuristic is useful or how to leverage them.
> > - How does Figure 1 in the revised draft compare to Table 1 in the original draft? In particular, I'm trying to understand the global accuracy curve. If I compute the average of the minimal per-task accuracy from Table 1 (which I imagine is the final accuracy) I obtain 55.46%, and if I compute it for the maximal per-task accuracy I obtain 56.38%. Both of these values are considerably below the final overall accuracy reported in the new Figure 1, which is roughly 66%. I don't understand how this value could be above the average of the peak per-task accuracies.
> > - The authors mention that that new tasks' accuracy can be improved by increasing the number of epochs or increasing the number of units. Was either of these hypotheses tested? Since we aren't shown learning curves, how can we know whether the algorithm had already converged for each new task? Also, if we just increase the number of epochs, wouldn't that similarly increase the amount of forgetting?
> > - I thank the authors for the added clarifications in Sections 2 and 3.1. I believe they help guide the reader better. There's still some work towards making the writing clearer, like further detailing how reducing correlation helps in later sections, but this is a good improvement.
> > - While I agree that state-of-the-art performance is neither a necessary nor a sufficient condition for scientific publication, I am left to wonder _what_ the contribution of the paper is. The technique certainly is novel, which is good, but what are its advantages? The authors mention scalability, but again, the paper states that scalability comes from manually adding units to the network at some unspecified time. Beyond that, it doesn't seem to have clear advantages for performance, as evidenced by the mixed results in Table 1 in the original draft and by the lack of comparison to stronger techniques like experience replay, or in terms of providing new insight. For me, the main contribution of the system is to use hashing to select routing paths. The effect of this (e.g., comparing it to other routing techniques like those in Rosenbaum et al.) is not analyzed.
> > - Stable features: it is very clear how to leverage a feature vector for both reinforcement and unsupervised learning. All the baselines in the paper could also be used in this way, and in fact at least EWC has been used in reinforcement learning. What is unclear is how the proposed approach creates _stable_ features that could help improve learning performance in those settings. Also, the authors still don't really explain what stable features are. Without any evidence of applicability to reinforcement or unsupervised learning, I don't believe this claim is relevant.
> > - Section 3.2.2: Thanks for the clarification regarding the use of residuals. I believe adding this to the draft could help understand the rationale better.
> > - I have two issues with the residual sparsity. First, Table 1 in Appendix B seems to suggest quite the opposite: making the residual regularization $\lambda=0$ has no negative effect, maintaining Task 0's final accuracy unchanged but increasing peak per-task accuracy. Second, ablating its effect does not give us any intuition about why we would want to regularize the residuals. The authors still don't explain what motivates this choice, which at first glance seems odd: the feature vectors, rather than the weights on them, are encouraged to be sparse.
> > - Section 3.4: I appreciate the addition of a suggestion for how to automatically add new units. I encourage the authors to explore this direction further in a future iteration of this work, if it isn't accepted for publication in its current version. As it stands, it is quite unclear if this would work well, since it isn't empirically tested.
> >
> >
> > Overall, the authors provide responses to all of my four main points, but only convinced me that one of them (clarity of the manuscript) was improved. I don't believe that significantly changes the quality of the work, and so my initial assessment still stands.

---

### Official Review · AnonReviewer4 · 2020-10-29
**Official Blind Review #4**

**Rating:** 6
**Confidence:** 3

**Review:**

Summary:
For a learning model to learn continuously, it needs to handle new datasets without catastrophic forgetting or requiring the model to grow larger. This paper proposes a hash-routed convolution neural network where a different set of convolution filters are used depending on the data. New convolution filters can be added to the network without increasing the computational cost of the network.

Questions:
1) What is the running time performance for hash-routed networks (HRN) during training and inference?

2) How are new classes added to the Softmax layer during training?

3) Couldn't you train a single model on the old and new datasets such that it performs well on both? If so, does the continual learning model train faster than building a data-centric model?

---

> ### Author Response · Authors · 2020-11-19
> **R4 specific comments**
>
> 1- Runtime performance: Please see the updated Appendix for runtime performance.
> 2- New classes management: HRN generate stable feature vectors and classification is performed using a task specific MLP classifier. There is therefore no need to adapt the softmax layer. Please refer to section 5.1 for further detail.
> 3- Training a single model on a combination of the old and new datasets will lead to good performance on both datasets. However, in a continual learning context, data is only available sequentially hence, such a combination is not usually possible.

---

### Official Review · AnonReviewer2 · 2020-10-29
**Ok but not good enough. Conceptually and empirically not entirely convincing.**

**Rating:** 4
**Confidence:** 4

**Review:**


### Summary

Ok, but not good enough. The authors present Hash-Routed Convolutional Neural Networks (HRNs), intended to enable learning of stable representations for continual learning, i.e. representations that change little for previously-learned tasks as more tasks are learned. The authors benchmark HRNs against several baselines in a number of tasks. Overall, the proposed method does not appear as conceptually and empirically convincing as those of other papers at ICLR and similar conferences.


### Reasons for score

- The presented algorithm does not appear as conceptually compelling as existing work on the subject (see below for papers that the authors could have cited and evaluated against, but did not).
- Overall, the architecture, and in particular the use of hash routing as it is used here, does not seem natural to me. (See explanation below)
- The results from the evaluation section are mixed, reinforcing my intuition that the proposed architecture is not as suitable as existing solutions.
- The paper omits information about hyper-parameters, hyper-parameter tuning and choices, making the paper’s results impossible to reproduce. Overall, there is so little information that it is impossible to tell if the experimental evaluation was fair to existing algorithms.


### Pros

- The paper addresses an important question: how can we train models in a continual-learning set-up, build representations that are meaningfully shared across successive tasks, and avoid performance degradation on previously learned tasks as new tasks are learned? The authors attempt to solve this problem using feature hashing, which had not previously been used in this particular way before. The proposed architecture is novel.

### Cons

- Conceptually, I find the HRN algorithm presented here somewhat unnatural. The authors write “Similar feature maps get to be processed by the same units, as a consequence of using feature hashing for routing. ” That’s technically true, but since the similarity is measured right in the raw feature space, the similarity will often not be meaningful. For example, a black dog in front of a green background would have little measured pixel-level similarity with a beige dog in front of a blue background, even though conceptually both images contain dogs. I would expect that first computing higher-level representations and then using those to route to an appropriate CNN would be more useful than measuring similarity on raw inputs. I do understand that the HRN inference proceeds through multiple iterations, and that the output of the first iteration becomes an input to the second iteration. Still, I imagine that hashing raw inputs right at the start to decide where to route is  harmful. Note: while Weinberger et al (https://alex.smola.org/papers/2009/Weinbergeretal09.pdf) did hash their input features, but they were using bag of words features, not pixel features.
- Figure 2 (top) is misleading and cherry-picks data that make the author’s proposed algorithm look better. The graph suggests (green line) that the authors’ HRN algorithm’s performance on Task 0 stays nearly constant at a high level as more tasks are added. However, a closer look at Table 1 (top) indicates that HRN performs significantly worse than baselines on several tasks (see e.g. task T1, T2 and T3), both in max and min accuracy.

- The paper does not provide (enough/any) information on the precise architectures and hyperparameters that were used for the “Experiments” section. This makes the paper very challenging to reproduce. Even for the HRN algorithm presented in this paper, I could not find what value was used for the maximum basis size parameter “m” was used. I could not find much information on whether the authors selected or tuned the hyper parameters for the algorithms that they benchmarked against in the “Experiments” section. Based on the information provided, I cannot tell if the experiments made were fair to the baselines.

- Some relevant existing work has not been cited and compared to: the authors did not cite the ANML algorithm (https://arxiv.org/pdf/2002.09571.pdf), which has achieved state-of-the-art performance on tasks similar to the one that the papers of the paper reviewed here evaluate on. I would suggest that the authors cite this paper, and benchmark againnnst ANML unless there is a good reason not to do so.

- The authors should also have cited “Routing Networks: Adaptive Selection of Non-Linear Functions for Multi-Task Learning” (ICLR 2018, https://openreview.net/forum?id=ry8dvM-R-) .

- The authors write ' *The output of a typical CNN is a feature map with a dimension that depends on the number of output channels used in each convolutional layer. In a HRN, this would lead to a variable dimension output as the final feature map depends on the routing.* ' Why would there necessarily be a variable-length output dimension? One could fix the output width of multiple CNNs, and sum the outputs element-wise to get a fixed-length representation.

- the authors write “As the network trains, hashed features will also change and routing might need adjustment. If nothing is done to update full basis, the network might get ”stuck” in a bad configuration. ” Would these basis updates not lead to changing output representations, thus breaking one of the key properties that the authors were looking for in this architecture? In fact, in Task T1 Table 1 (top), it appears that the model’s performance degraded quite significantly as tasks were learned.

### Questions during rebuttal period:

- I would be grateful if the authors could address as many of the cons listed above as possible.
- In Figure 4, on the right, there are arrows going into U_4, but no arrows coming out of U_4. Is this an error?

### Some suggestions:

- There are possessive apostrophes missing here and there, e.g. “units” vs “unit’s”. I suggest that the authors review their use of possessive apostrophes one more time.

- “0 This is a 11 tasks” should be “0 This is an 11 tasks”

- “The following scenarios were considered.” might better be followed by a colon than a period.

---

> ### Author Response · Authors · 2020-11-19
> **R2 specific comments**
>
> - Similarity: It is true that similarity in the raw pixel space might not always be meaningful and that this might lead to performance degradation. This is particularly true for large images (e.g. 64x64 pixels or more) but we found that performance was acceptable for small patches (e.g. 32x32 or less). Using a first stage feature generator such as a pre-trained convolution layer with frozen weights is actually something we had considered. It is an interesting suggestion but we chose not to discuss it in this paper for two main reasons. First, it does not change the nature of our algorithm as the output of the primary layer can be seen as a raw image (even though it has more structure). Second, it limits our network's scalability as the primary layer is not necessarily suited to handle all types of images. Otherwise, it would need to be a large pre-trained network that would significantly increase the computational cost of the HRN.
> - Figure 2: Figure 2 plots have been updated and additional plots were added to show top task accuracy and accuracy drop.
> - Precise architecture and hyperparameters: Please see the updated Appendix. The supplementary material contains the full code and configuration files to reproduce all experiments.
> - ANML paper: we have now mentioned this paper in the related work section but we do not consider the chosen approach to be of interest for comparison to our approach (see the discussion in the updated version of the paper). Indeed, we do not claim to surpass state-of-the-art continual learning algorithms with HRN. We only present a new scalable approach with good results on typical benchmarks.
> - ICLR 2018 paper: we have now mentioned this very interesting paper in the related work section (dynamic networks).
> - Obtaining fixed-length representations: R2's suggestion is again something we had tried at first. This did not lead to acceptable performance and this is what prompted the search for fixed length representations that eventually led to the use of feature hashing.
> - Basis update: The basis update algorithm only updates the least stable part of the basis, keeping as much stability as possible in the output representation.
> - Figure 4: The figure is correct, there is no missing arrow that should be coming out of U_4. Indeed, when the final unit is selected (maximum depth has been reached), there is no need to perform the final convolution. Only the projection onto the final unit's basis is needed, to compute the last residue vector.

---

> > ### Comment · AnonReviewer2 · 2020-11-24
> > **R2 comment following authors' response**
> >
> > - The authors' revisions made the paper more clear, and the added information from the appendix makes the work more reproducible.
> > - Overall, my assessment of the paper's key ideas has not changed, so I did not change my review score. The paper's ideas and proposed architecture still do not seem natural to me, and the empirical results results by themselves are not strong enough for publication at ICLR.

---

### Author Response · Authors · 2020-11-19
**Global comment**

We would like to thank the reviewers for their positive feedback and valuable comments. We hope the following answers will address the reviewers' concerns. We have tried to address as many questions and issues as possible, through an updated version of the paper and the supplementary material.
In brief, here are the main changes:

Paper:
Typos and other suggested formatting corrections were made. Several clarifications were added. Related work section was updated. Global accuracy plots were added to existing plots. Two additional plots were added to SVHN/incremental-Cifar100 scenario for a better analysis, removing the need for the large accuracy scores table. The appendix now includes many implementation details, detailed ablation and runtime performance.


Supplementary material:
Code has been updated to latest pytorch version and several optimizations were implemented, improving overall runtime. Ablation experiments configuration files are more explicit and have been put into a dedicated folder. EWC now has a dedicated classifier per task (as suggested by R3) and results were updated accordingly.

Specific concerns and questions will be adressed separately to each reviewer.

---

### Decision · Program_Chairs · 2021-01-07
**Final Decision**

**Decision:**

Reject

**Comment:**

Reviewers raised several concerns about the paper guided by unfounded heuristics as well as the artificiality of the tasks involved.  Rebuttal only answered a few of them and did not convince the reviewers which has been clearly stated in the response. We hope that the authors will improve the paper for future submission based on the reviews.